# Association between Immune Checkpoint Inhibitor Treatment Outcomes and Body Composition Factors in Metastatic Renal Cell Carcinoma Patients

**DOI:** 10.3390/cancers15235591

**Published:** 2023-11-26

**Authors:** Kohei Takei, Toshiki Kijima, Naoya Okubo, Ryo Kurashina, Hidetoshi Kokubun, Toshitaka Uematsu, Hironori Betsunoh, Masahiro Yashi, Takao Kamai

**Affiliations:** Department of Urology, Dokkyo Medical University, 880 Kitakobayashi, Mibu, Shimotsuga 321-0293, Tochigi, Japan; takei-74@dokkyomed.ac.jp (K.T.); kubonao@dokkyomed.ac.jp (N.O.); k-ryo@dokkyomed.ac.jp (R.K.); hide-k@dokkyomed.ac.jp (H.K.); tuematsu@dokkyomed.ac.jp (T.U.); hirobets@dokkyomed.ac.jp (H.B.); yashima@dokkyomed.ac.jp (M.Y.); kamait@dokkyomed.ac.jp (T.K.)

**Keywords:** body composition-related biomarker, immune checkpoint inhibitors, renal cell carcinoma, sarcopenia

## Abstract

**Simple Summary:**

Although body composition-related biomarkers are associated with the prognosis of patients with cancer, whether they are associated with the therapeutic effects of immune checkpoint inhibitors remains unclear. We found that among body composition-related biomarkers, sarcopenia based on skeletal muscle mass was strongly associated with the treatment efficacy of immune checkpoint inhibitors in patients with metastatic renal cell carcinoma. However, body composition-related biomarkers based on subcutaneous or visceral fat were not associated with treatment efficacy. The skeletal muscle releases myokines to activate the immune system. Therapeutic interventions for sarcopenia may not only improve patients’ quality of life but also improve the therapeutic effects of immune checkpoint inhibitors and prolong the prognosis of patients.

**Abstract:**

Introduction: Immune checkpoint inhibitors (ICIs) have revolutionized the treatment of metastatic renal cell carcinoma (mRCC); however, validating body composition-related biomarkers for their efficacy remains incomplete. We evaluated the association between body composition-related markers and the prognosis of patients with mRCC who received ICI-based first-line therapies. Patients and Methods: We retrospectively investigated 60 patients with mRCC who underwent ICI-based therapy as their first-line treatment between 2019 and 2023. Body composition variables, including skeletal muscle, subcutaneous fat, and visceral fat indices, were calculated using baseline computed tomography scans. Sarcopenia was defined according to sex-specific cut-off values of the skeletal mass index. The associations between body composition indices and objective response rate (ORR), disease control rate (DCR), progression-free survival (PFS), and overall survival (OS) were evaluated. Results: Patients with sarcopenia had lower ORR and DCR than those without sarcopenia (33.3% vs. 61.1%, *p* = 0.0436 and 52.4% vs. 94.4%, *p* = 0.0024, respectively). Patients with sarcopenia had a significantly shorter median PFS (14 months vs. not reached, *p* = 0.0020) and OS (21 months vs. not reached, *p* = 0.0023) than patients without sarcopenia did. Sarcopenia was a significant predictor of PFS (hazard ratio [HR], 4.31; 95% confidence interval [CI], 1.65–14.8; *p* = 0.0018) and OS (HR, 5.44; 95% CI, 1.83–23.4; *p* = 0.0013) along with poor IMDC risk. No association was found between the subcutaneous, visceral, and total fat indices and the therapeutic effect of ICI-based therapy. Conclusions: Sarcopenia was associated with a lower response and shorter survival rates in patients with mRCC who received first-line ICI-based therapy.

## 1. Introduction

Immune checkpoint inhibitors (ICIs), either alone or in combination with tyrosine kinase inhibitors (TKIs), provide improved outcomes for patients with metastatic renal cell carcinoma (mRCC) and have become the standard first-line treatment in recent years [1]. Nivolumab, a programmed cell death protein-1 (PD-1) inhibitor, was the first approved ICI for mRCC [2]. Additionally, the combination of nivolumab and ipilimumab, a cytotoxic T-lymphocyte-associated protein 4 inhibitor, was approved as a first-line treatment option because of its potential to provide a high complete response (CR) rate and long-term survival benefit [3]. The approval of various combinations of ICIs and TKIs has led to a growing utilization of ICI-based therapies in patients with mRCC. However, not all patients respond to these treatments. Therefore, clinical biomarkers are needed to predict which patients will benefit from ICI-based therapies. Despite extensive research into potential biomarkers via blood tests [4], pathological analyses [5], and genetic testing of tumor tissue [6], robust biomarkers have not yet been established. There is a need for clinically available biomarkers based on information derived before ICI administration in routine clinical practice.

The prognostic role of body mass index (BMI), a body composition-based index, has been studied extensively in multiple types of cancers, including RCC. Although patients with a high BMI have a higher risk of developing RCC, these patients tend to have a better prognosis, a phenomenon often referred to as the “obesity paradox” [7]. Although the relationship between BMI and prognosis in patients with mRCC receiving ICIs is inconclusive, a recent study reported a better prognosis in patients with high BMI compared with normal-weight patients [8]. BMI is simple and easy to calculate; however, it is an imperfect index that treats all body composition components equally, which can lead to significant heterogeneity in skeletal muscle and fat mass even among patients with identical BMI. Therefore, detailed body composition measurements, such as muscle and fat masses, are gaining attention. The standard method for measuring body composition involves measuring the areas of skeletal muscle and subcutaneous and visceral fat components based on computed tomography (CT) scans of the height of the third lumbar vertebra. These body composition indices have been reported to be associated with survival, perioperative complications, and adverse events of chemotherapy in some types of cancer [9,10].

Several studies have examined the association between these body composition markers and treatment efficacy in patients with mRCC undergoing ICI-based therapy. Martini et al. evaluated fat and skeletal muscle mass as well as intramuscular fat (myosteatosis) in patients with mRCC receiving ICI-based therapy and reported that patients with high subcutaneous and visceral fat and less myosteatosis had a better prognosis than those counterpart [11]. Ged et al. evaluated skeletal muscle, subcutaneous fat, and visceral fat mass, showing that patients with high muscle mass had a better prognosis than those with low muscle mass [12]. Recently, McManus and colleagues reported that the prognosis was better in patients with low skeletal muscle mass and subcutaneous fat mass, contrary to the results from previous reports [13]. Thus, further study is warranted to evaluate these inconclusive findings regarding the association between body composition markers and the treatment efficacy of ICI-based therapy in patients with mRCC. Therefore, this study aimed to investigate the association between body composition markers and other clinical factors and clinical outcomes in patients with mRCC who received ICI-based therapy as first-line therapy.

## 2. Patients and Methods

### 2.1. Patients

We retrospectively investigated 60 patients with mRCC who received ICI-based therapy between 2019 and 2023 at Dokkyo Medical University Hospital. All patients were clinically diagnosed with mRCC and received ICI-based first-line therapy. All patients underwent an analyzable abdominal CT scan within 30 days of the start of ICI-based therapy. The following clinical information was obtained from medical records at the beginning of ICI therapy: age, sex, weight, height, prior nephrectomy, Eastern Cooperative Oncology Group performance status (ECOG PS), tumor histology, International Metastatic RCC Database Consortium (IMDC) risk classification, site of metastasis, neutrophil-lymphocyte ratio (NLR), platelet-lymphocyte ratio (PLR), and C-reactive protein. The occurrence of immune-related Adverse Events (irAEs) was assessed using medical records and graded according to the Common Terminology Criteria for Adverse Events version 5.0.

### 2.2. Body Composition Analysis

The BMI at the initiation of ICI-based therapy was calculated using the standard BMI formula (kg/m^2^) = [weight in kg]/[height in m]^2^. The patients were classified into normal weight (BMI: 18.5 to <25 kg/m^2^), overweight (BMI: 25 to < 30 kg/m^2^), and obese (BMI ≥30 kg/m^2^) according to the World Health Organization (WHO) categories for BMI.

Body composition parameters, including skeletal muscle area, subcutaneous fat area, and visceral fat area, were meticulously assessed by a single urologist, Dr. K. Takei, employing a state-of-the-art 3-dimensional image analysis system (SYNAPSE VINCENT; FUJIFILM Healthcare Corporation, Tokyo, Japan) (Figure 1). The third lumbar vertebra (L3) was used as the landmark to measure these parameters. In assessing the skeletal muscle area, we precisely identified the cross-sectional area of major skeletal muscle groups, including the rectus abdominis, internal and external obliques, lateral obliques, psoas, quadratus lumborum, and erector spinae. Hounsfield unit (HU) thresholds were used to differentiate tissues: −190 to −30 HU for subcutaneous fat tissue, −150 to −50 HU for visceral fat tissue, and −29 to 150 HU for skeletal muscle. The skeletal muscle index (SMI) was calculated as the skeletal muscle area at L3 divided by the square of height. The subcutaneous fat index (SFI) and visceral fat index (VFI) were similarly calculated as the subcutaneous fat area and the visceral fat area divided by the square of the height. Total fat index (TFI) was calculated as the sum of SFI and VFI.

Sarcopenia was precisely defined using sex-specific cut-off values for SMI, as outlined in the study by Martin et al. These cut-offs were determined as follows: for men with a BMI < 25 kg/m^2^, an SMI < 43 cm^2^/m^2^ was indicative of sarcopenia; for men with a BMI ≥ 25 kg/m^2^, an SMI < 53 cm^2^/m^2^ was considered indicative; and for women, an SMI < 41 cm^2^/m^2^ was used to define sarcopenia [14]. As no validated cut-off exists for SFI, VFI, and TFI, they were classified into high and low using sex-specific median values as cut-off, similar to previous studies [12,13].

### 2.3. Statistical Analyses

Overall survival (OS) and progression-free survival (PFS) were calculated from the start of the ICI-based therapy for each event. The objective response rate (ORR) was the proportion of patients who achieved CR or partial response (PR). The disease control rate (DCR) was defined as the proportion of patients who achieved CR, PR, or stable disease (SD) for >3 months according to the RECIST guidelines.

Continuous variables were reported as median/interquartile range, and categorical variables were reported as number (percentage). Categorical variables were analyzed using Fisher’s exact test, and continuous variables were analyzed using the Mann–Whitney U test. PFS and OS were estimated using the Kaplan–Meier survival curve method and were compared using the log-rank test. Univariate and multivariate analyses with Cox proportional hazards regression models were used to identify the factors related to survival. All statistical analyses were performed using JMP version 13 (SAS Institute, Cary, NC, USA), and statistical significance was defined as *p* < 0.05.

## 3. Results

### 3.1. Patient Characteristics

The clinical characteristics of the 60 patients are presented in Table 1. The median age was 71 years, and 46 patients (77%) were men. Thirty-four (57%) patients received ICI combination therapy (ipilimumab–nivolumab), and the remaining twenty-six (43%) received ICI-TKI combination therapy (avelumab–axitinib in 12, pembrolizumab–axitinib in 5, pembrolizumab–envatinib in 5, and nivolumab–cabozantinib in 4). Prior nephrectomy was performed in 27 (45%) patients, and 38 (63%) had good PS (≤1). Forty patients (67%) had clinical T stage 3 or greater, and forty-four (73%) had clear cell histology. The IMDC risk was intermediate in 38 (63%) patients, whereas the remaining 22 (37%) had poor risk. The median BMI and SMI values were 22.9 kg/m^2^ and 41.1 cm^2^/m^2^, respectively. Sex-specific median values of SFI, VFI, and TFI were as follows: (SFI: 28.7 cm^2^/m^2^ in men, 60.5 cm^2^/m^2^ in women; VFI: 44.1 cm^2^/m^2^ in men, 27.8 cm^2^/m^2^ in women: TFI: 77.3 cm^2^/m^2^ in men, 89.5 cm^2^/m^2^ in women).

According to the WHO criteria, 16 patients were categorized as overweight, and the remaining 44 were classified as having average weights. According to the criteria described by Martin et al. [14], 42 patients were identified as having sarcopenia. Patients with sarcopenia had more non-clear cell histology, higher NLR, higher PLR, lower BMI, lower SMI, lower VFI, and lower TFI (*p* = 0.02) than those without sarcopenia (Table 2).

### 3.2. Association between Body Composition Markers and Objective Responses

The ORR and DCR of the 60 patients were 41.6% and 65%, respectively. Both patients with and without sarcopenia achieved CR, PR, and SD; however, patients who developed progressive disease (PD) were mainly found to have sarcopenia. Patients with sarcopenia had a lower ORR and DCR than those without sarcopenia (33.3% vs. 61.1%, *p* = 0.0436 and 52.4% vs. 94.4%, *p* = 0.0024, respectively). BMI, SFI, VFI, and TFI were not correlated with ORR and DCR.

### 3.3. Association between Body Composition Markers and Survival Outcomes

During follow-up (median, 15 months; range, 1–52 months), 30 (50%) patients developed PD, and 26 (43%) died of the disease. Patients with sarcopenia had significantly shorter median PFS (14 months vs. not reached, *p* = 0.0020) and OS (21 months vs. not reached, *p* = 0.0023) compared with patients without sarcopenia (Figure 2). The negative impact of sarcopenia on PFS and OS was confirmed in a subgroup analysis of patients receiving ICIs and ICI-TKI therapy. BMI, SFI, VFI, and TFI were not correlated with PFS or OS.

Table 3 shows the Cox regression analysis for PFS. In univariate analysis, age, IMDC risk, NLR, PLR, and sarcopenia were associated with PFS. In multivariate analysis, sarcopenia was a significant predictor of PFS (hazard ratio [HR], 4.31; 95% confidence interval [CI], 1.65–14.8; *p* = 0.0018) along with poor IMDC risk (HR, 2.61; 95% CI, 1.22–5.67; *p* = 0.0136). In univariate Cox regression analysis for OS (Table 4), IMDC risk, NLR, and sarcopenia were associated with OS. In multivariate analysis, sarcopenia was a significant predictor of OS (HR, 5.44; 95% CI, 1.83–23.4; *p* = 0.0013) along with poor IMDC risk (HR, 3.30; 95% CI, 1.48–7.76; *p* = 0.0037).

### 3.4. Association between Body Composition Markers and irAEs

During follow-up, 34 patients experienced irAEs of all grades, and 21 had irAEs ≥ grade 3 (Table 5). The frequently observed irAEs were as follows: hepatic dysfunction (*n* = 5), adrenal insufficiency (*n* = 4), interstitial pneumonia (*n* = 3), rheumatoid arthritis (*n* = 3), thyroiditis (*n* = 2), hypophysitis (*n* = 2). irAEs of all grades were more common in patients without sarcopenia than in those with sarcopenia (78% vs. 48%; *p* = 0.0463). However, the frequency of severe (≥grade 3) irAEs was not associated with the presence of sarcopenia. The BMI, SFI, VFI, and TFI were not associated with the frequency of all-grade irAEs. Treatment discontinuation due to irAEs occurred in 13 patients. The frequency of treatment discontinuation due to irAEs was eight (19%) in patients with sarcopenia and five (27%) in patients without sarcopenia, with no significant difference (*p* = 0.503).

## 4. Discussion

Numerous studies have reported that body composition markers affect treatment outcomes, prognosis, and adverse events in various cancers, including mRCC. Among patients with mRCC receiving ICI-based therapy, several studies have reported a better prognosis in individuals with higher BMI and muscle and fat mass; however, these results are still inconclusive [11,12,13]. In this study, we investigated whether body composition markers correlate with the treatment efficacy and survival outcomes of patients administered ICI-based therapy as first-line therapy. Our results showed that the ORR and DCR were significantly lower in patients with sarcopenia than in those without it, and patients who developed PD were primarily those in the sarcopenia group. Additionally, patients with sarcopenia had a significantly shorter OS and PFS than those without sarcopenia. Furthermore, multivariate analysis revealed that sarcopenia was an independent prognostic factor along with poor IMDC risk. None of the indices related to fat components demonstrated a significant correlation with the treatment efficacy of ICI-based therapy. These results suggest that sarcopenia negatively affects treatment efficacy and prognosis in patients with mRCC treated with ICI-based therapy.

These findings suggested a possible association between sarcopenia and an immunosuppressive microenvironment. One of the mechanisms by which sarcopenia causes immunosuppression is a decrease in myokines [15]. Skeletal muscles have recently been recognized as endocrine organs capable of secreting more than 600 molecules, termed myokines [16]. Skeletal muscle can modulate the tumor microenvironment by secreting these myokines. One of the myokines that affect the tumor microenvironment is interleukin (IL)-15, which is abundantly expressed in skeletal muscle and activates natural killer (NK) and T cell immunity [17]. IL-15 has been found to co-localize within the tumor microenvironment with cluster of differentiation (CD)8+ T and NK cells [18], and the injection of IL-15 suppressed tumor burden by increasing intratumoral CD8+ T and NK cells in a mouse model [19]. To enhance the therapeutic efficacy of ICIs, recombinant human IL-15 or its agonists have been investigated in clinical trials for leukemia [20], non-small cell lung, [21], and ovarian cancers [22]. In addition to the administration of IL-15, exercise-induced increases in IL-15 were reported to enhance CD8+ T cell immunity and suppress tumor growth in mouse models. These findings suggest that IL-15 secreted from skeletal muscle promotes anti-tumor immunity and that patients with sarcopenia may have impaired immunity due to decreased IL-15.

Another myokine that may have anti-cancer potential is decorin, ubiquitously expressed in connective tissues [23]. Decorin acts as a soluble inhibitor of pan-receptor tyrosine kinases, such as the Met receptor and vascular endothelial growth factor receptor [24]. As these receptor tyrosine kinases have essential roles in the development and progression of RCC, and TKIs for these receptors are used in the treatment of mRCC, skeletal muscle-derived decorin might have anti-tumor effects in mRCC.

Chronic inflammation is another factor that may explain the link between sarcopenia and the immunosuppressive microenvironment. Chronic inflammation induces myeloid-derived suppressor cells, which suppress the activity of CD8+ T-cells and activate immunosuppressive regulatory T-cells [25,26,27]. Additionally, the inflammatory cytokines, including tumor necrosis factor-alpha (TNF-α), IL-1, and IL-8, affect skeletal muscle, leading to cancer cachexia [28]. TNF-α activates the nuclear factor kappa-light-chain-enhancer of activated B cell pathway and ubiquitin-mediated proteasome catabolism of muscle protein [29]. Therefore, chronic inflammation in patients with sarcopenia suppresses anti-tumor immunity, resulting in a significantly poorer response to ICI therapy [30].

Several previous studies have suggested a possible association between fat components and the therapeutic effects of ICIs; however, this was not confirmed in the current study. The fact that adipocytes express PD-ligand 1 (L1) during adipogenesis suggests that adiposity promotes tumor immune evasion, which ICIs could reverse [31]. Furthermore, adipose tissue is a secondary lymphoid organ that harbors numerous immune cells, including T cells [32], suggesting a better response to ICIs in patients with a higher composition of fat tissue. The relationship between adipose tissue, the immune microenvironment, and the therapeutic effects of ICIs requires further investigation.

In addition to the influence of skeletal muscle and fat components on anti-tumor immunity, the treatment of comorbidities may also impact the prognosis of patients with mRCC. Previous studies have reported that antidiabetic [33,34] and antihypertensive drugs [35] have potential anti-tumor effects on RCC and that patients who take these drugs tend to have better prognoses than those who do not. Although this study did not specifically focus on medications for comorbidities, further investigation is needed to evaluate the effects of antidiabetic and antihypertensive drugs on the prognosis of patients with mRCC receiving ICI-based therapy.

For a cancer patient with sarcopenia, improving appetite and weight not only improves the general condition and quality of life but may also improve the anti-tumor immune status and lead to an improved prognosis. Thus, therapeutic intervention against sarcopenia is warranted for patients with cancer. Therapeutic interventions for sarcopenia in patients with cancer may include counseling by dietitians, medications, and exercise [36]. Counseling by dietitians has been reported to improve the weight of cancer patients with sarcopenia [37]. For drug therapy, progesterone analogs and corticosteroids are weakly recommended. Progesterone analogs have been reported to mildly improve appetite, weight, and quality of life in cancer patients with sarcopenia [38]. Corticosteroids have also been reported to improve appetite in patients with cancer; however, there have been no reports showing an improvement in weight with its administration [39]. The appropriate dose or duration of treatment for these drugs has not been determined, and they should be used with caution due to the possibility of adverse events. Although there is little evidence regarding the impact of exercise on patients with cancer, exercise is currently being evaluated as part of multimodal interventions for these patients [40].

In the current study, the occurrence rate of irAEs of all grades was higher in patients without sarcopenia than in those with it. However, the frequency of severe irAEs (>grade 3) was comparable among patients with and without sarcopenia. These results suggest that patients without sarcopenia are more susceptible to irAEs and more likely to benefit from ICI-based therapy. This is consistent with previous reports revealing that patients who experience irAEs exhibit higher response rates and more prolonged survival with ICI-based therapy [41]. The mechanism of irAEs is thought to be an autoimmune mechanism in which T cells activated by ICIs attack autologous normal tissue. Not only does anti-tumor immunity come into play, but autoimmunity against normal tissues can also be activated by ICIs, which may explain why patients with irAEs had a better response to ICIs. The association between sarcopenia and irAEs requires further investigation.

This study has some limitations. First, this was a retrospective single-center study with a small sample size and a short follow-up period. Thus, large-scale, multicenter, prospective studies are necessary to improve the accuracy of findings. Second, the criteria for sarcopenia were calculated using data from a Canadian cohort, which may be inappropriate for Asians. Third, sarcopenia should be evaluated using different methods, such as indices focusing on specific muscles and their longitudinal changes over time. Some studies have reported that sarcopenia based on the psoas muscle index was a significant prognostic factor in patients with RCC who underwent nivolumab therapy [42]. In another study, the rate of change of the psoas major muscle area was useful in patients with non-small cell lung cancer treated with nivolumab or pembrolizumab [43]. In addition to body composition-related markers, other factors relevant to sarcopenia, such as muscle strength and exercise capacity, should also be assessed in patients receiving ICI-based therapy. Finally, this study included patients receiving both the ICI and the ICI-TKI combinations. Because the effects of sarcopenia on treatment efficacy may differ between ICI and ICI-TKI combinations, further studies focusing on the single regimen are warranted.

## 5. Conclusions

In conclusion, sarcopenia was associated with lower response rates and shorter survival in patients with mRCC who received ICI-based therapy as first-line treatment. No association was found between the fat components and the therapeutic effects of ICI-based therapy. Furthermore, body composition markers can predict the prognosis of patients with mRCC receiving ICIs. However, body composition markers measured using different indices and longitudinal changes over time require further investigation.

## Figures and Tables

**Figure 1 cancers-15-05591-f001:**
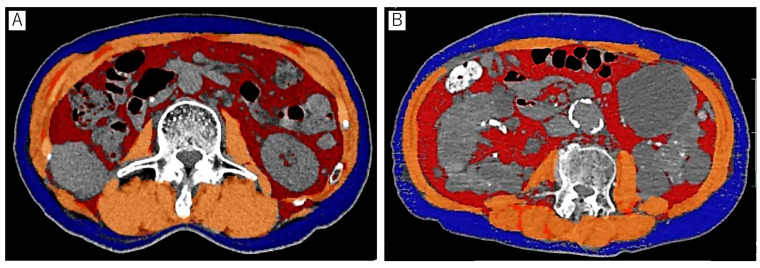
Measurement of skeletal muscle (orange), subcutaneous fat (blue), and visceral fat component (red) using SYNAPSE VINCENT. Representative computed tomography images of a patient without sarcopenia (**A**) and a patient with sarcopenia (**B**).

**Figure 2 cancers-15-05591-f002:**
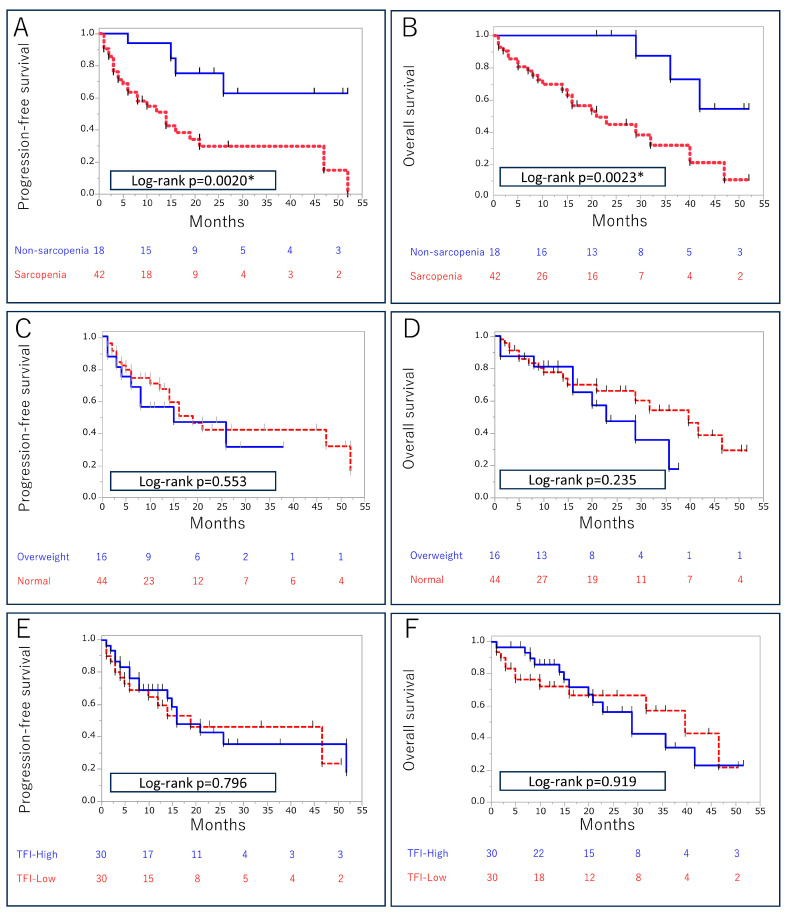
Progression-free survival and overall survival after ICI-based therapy in relation to sarcopenia (**A**,**B**), body mass index (**C**,**D**), and total fat index (**E**,**F**). Asterisk (*) indicates statistical significance.

**Table 1 cancers-15-05591-t001:** Patient characteristics and body composition data at initiation of ICI-based therapy.

	Overall
	*n* = 60
Age (years), median (IQR)	71 (63–75)
Sex, *n* (%)	
Male	46 (77)
Female	14 (23)
ICI-based therapy, *n* (%)	
ICI combination	34 (57)
ICIs + TKIs	26 (43)
Prior nephrectomy, *n* (%)	
Yes	27 (45)
No	33 (55)
ECOG PS, *n* (%)	
≤1	38 (63)
≥2	22 (37)
Clinical T stage, *n* (%)	
≤2	20 (33)
≥3	40 (67)
Histology, *n* (%)	
Clear cell carcinoma	44 (73)
Non-clear cell carcinoma	10 (17)
Unknown	6 (10)
IMDC risk score, *n* (%)	
Intermediate	38 (63)
Poor	22 (37)
Site of metastasis, *n* (%)	
Lymph nodes	20 (33)
Lung	35 (58)
Liver	6 (10)
Other organs	28 (47)
NLR, median (IQR)	3.52 (2.29–4.84)
PLR, median (IQR)	188 (133–244)
CRP (mg/dL), median (IQR)	0.81 (0.08–5.96)
BMI (kg/m^2^), median (IQR)	22.9 (20.9–25.2)
SMI (kg/m^2^), median (IQR)	41.1 (37.1–49.6)
SFI (kg/m^2^), median (IQR)	31.7 (20.7–51.3)
VFI (kg/m^2^), median (IQR)	36.1 (19.1–56.8)
TFI (kg/m^2^), median (IQR)	80.2 (42.7–103)

Abbreviations: IQR, interquartile range; IMDC, International Metastatic RCC Database Consortium; ECOG, Eastern Cooperative Oncology Group; NLR, neutrophil-to-lymphocyte ratio; PLR, platelet-to-lymphocyte ratio; CRP, C reactive protein; BMI, body mass index; SMI, skeletal muscle index; SFI, subcutaneous fat index; VFI, visceral fat index; TFI, total fat index.

**Table 2 cancers-15-05591-t002:** Patient characteristics according to the presence of sarcopenia.

	Sarcopenia	Non-Sarcopenia	*p*-Value
	*n* = 42	*n* = 18	
Age (years), median (IQR)	72 (64–75)	69 (62–73)	0.428
Sex, *n* (%)			0.0455 *
Male	29 (69)	17 (94)	
Female	13 (31)	1 (6)	
ICI-based therapy, *n* (%)			0.779
ICI combination	23 (55)	11 (61)	
ICIs + TKIs	19 (45)	7 (39)	
Prior nephrectomy, *n* (%)			0.778
Yes	18 (43)	9 (50)	
No	24 (57)	9 (50)	
ECOG PS, *n* (%)			0.773
≤1	25 (60)	12 (67)	
≥2	17 (40)	6 (33)	
Clinical T stage, *n* (%)			0.766
≤2	15 (36)	5 (28)	
≥3	27 (64)	13 (72)	
Histology, *n* (%)			0.0417 *
Clear cell carcinoma	27 (64)	17 (94)	
Non-clear cell carcinoma	10 (24)	0	
Unknown	5 (12)	1(6)	
IMDC risk score, *n* (%)			0.155
Intermediate	24 (57)	14 (78)	
Poor	18 (43)	4 (22)	
Site of metastasis, *n* (%)			
Lymph nodes	14 (33)	6 (33)	1.000
Lung	25 (60)	10 (56)	0.783
Liver	5 (12)	1 (6)	0.658
Other organs	23 (55)	5 (28)	0.0893
NLR, median (IQR)	3.89 (2.78–5.64)	2.56 (1.93–3.86)	0.0209 *
PLR, median (IQR)	195 (145–307)	171 (112–196)	0.0330 *
CRP (mg/dL), median (IQR)	1.16 (0.06–6.19)	0.34 (0.14–2.79)	0.237
BMI (kg/m^2^), median (IQR)	22.7 (19.9–24.5)	24.4 (22.2–28.4)	0.0062 *
SMI (kg/m^2^), median (IQR)	39.3 (35.5–41.9)	51.4 (46.1–55.3)	<0.0001 *
SFI (kg/m^2^), median (IQR)	29.8 (14.5–51.6)	33.6 (25.7–51.5)	0.297
VFI (kg/m^2^), median (IQR)	30.8 (15.7–54.3)	53.2 (35.3–73.9)	0.0262 *
TFI (kg/m^2^), median (IQR)	72.7 (38.2–100)	93 (64.5–127)	0.0407 *

Abbreviations: IQR, interquartile range; IMDC, International Metastatic RCC Database Consortium; ECOG, Eastern Cooperative Oncology Group; NLR, neutrophil-to-lymphocyte ratio; PLR, platelet-to-lymphocyte ratio; CRP, C reactive protein; BMI, body mass index; SMI, skeletal muscle index; SFI, subcutaneous fat index; VFI, visceral fat index; TFI, total fat index. Asterisks (*) indicate statistical significance.

**Table 3 cancers-15-05591-t003:** Cox regression analysis of clinical and body composition variables for progression-free survival.

	Univariate Analysis		Multivariate Analysis	
Variable Category	HR (95% CI)	*p*-Value	HR (95% CI)	*p*-Value
Age (continuous)	0.96 (0.93–0.99)	0.0210 *		
Sex Female vs. male (ref)	1.83 (0.76–5.47)	0.191		
ECOG PS 2 vs. 0–1 (ref)	1.64 (0.77–3.41)	0.190		
IMDC risk Poor vs. intermediate (ref)	2.79 (1.32–6.00)	<0.0077 *	2.61 (1.22–5.67)	0.0136 *
Liver metastases Yes vs. no (ref)	2.25 (0.66–5.85)	0.175		
NLR (continuous)	1.20 (1.08–1.32)	0.0011 *		
PLR (continuous)	1.003 (1.000–1.004)	0.0275 *		
BMI Overweight vs. normal (ref)	0.79 (0.37–1.83)	0.566		
SFI Low vs. high (ref)	0.81 (0.39–1.66)	0.553		
VFI Low vs. high (ref)	0.87 (0.42–1.82)	0.717		
TFI Low vs. high (ref)	1.10 (0.52–2.29)	0.799		
Sarcopenia Yes vs. no (ref)	4.52 (1.75–15.4)	0.0010 *	4.31 (1.65–14.8)	0.0018 *

SFI, VFI, and TFI were categorized as high or low based on sex-specific median values as cut-offs. Asterisk (*) indicates statistical significance. Abbreviations: ECOG, Eastern Cooperative Oncology Group; IMDC, International Metastatic RCC Database Consortium; NLR, neutrophil-to-lymphocyte ratio; PLR, platelet-to-lymphocyte ratio; BMI, body mass index; SFI, subcutaneous fat index; VFI, visceral fat index; TFI, total fat index.

**Table 4 cancers-15-05591-t004:** Cox regression analysis of clinical and body composition variables for overall survival.

	Univariate Analysis		Multivariate Analysis	
Variable Category	HR (95% CI)	*p*-Value	HR (95% CI)	*p*-Value
Age (continuous)	0.97 (0.93–1.00)	0.0728		
Sex Female vs. male (ref)	1.03 (0.37–2.42)	0.953		
ECOG PS 2 vs. 0–1 (ref)	1.84 (0.82–4.05)	0.136		
IMDC risk Poor vs. intermediate (ref)	3.37 (1.52–7.80)	0.0029 *	3.30 (1.48–7.76)	0.0037 *
Liver metastases Yes vs. no (ref)	2.35 (0.68–6.27)	0.158		
NLR (continuous)	1.18 (1.05–1.30)	0.0058 *		
PLR (continuous)	1.002 (0.99–1.004)	0.0527		
BMI Overweight vs. normal (ref)	1.65 (0.68–3.38)	0.254		
SFI Low vs. high (ref)	1.05 (0.47–2.30)	0.899		
VFI Low vs. high (ref)	1.20 (0.55–2.65)	0.636		
TFI Low vs. high (ref)	1.04 (0.48–2.30)	0.919		
Sarcopenia Yes vs. no (ref)	5.45 (1.86–23.2)	0.0010 *	5.44 (1.83–23.4)	0.0013 *

SFI, VFI, and TFI were categorized as high or low based on sex-specific median values as cut-offs. Asterisk (*) indicates statistical significance. Abbreviations: ECOG, Eastern Cooperative Oncology Group; IMDC, International Metastatic RCC Database Consortium; NLR, neutrophil-to-lymphocyte ratio; PLR, platelet-to-lymphocyte ratio; BMI, body mass index; SFI, subcutaneous fat index; VFI, visceral fat index; TFI, total fat index.

**Table 5 cancers-15-05591-t005:** Association between body composition variables and immune-related adverse events.

	All-Grade irAEs, *n*(%)	*p*-Value	irAEs ≥ Grade 3, *n* (%)	*p*-Value
Sarcopenia				
Yes (*n* = 42)	20 (48%)	0.0463 *	12 (29%)	0.144
No (*n* = 18)	14 (78%)		9 (50%)	
BMI				
Overweight (*n* = 16)	9 (56%)	1.000	6 (38%)	1.000
Normal (*n* = 44)	25 (57%)		15 (34%)	
TFI				
High (*n* = 30)	18 (60%)	0.795	12 (40%)	0.589
Low (*n* = 30)	16 (53%)		9 (30%)	

Abbreviations: irAEs, immune-related adverse events; BMI, body mass index; TFI, total fat index. Asterisk (*) indicates statistical significance.

## Data Availability

The data presented in this study are available in this article.

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
