# Peer review of "Association between Immune Checkpoint Inhibitor Treatment Outcomes and Body Composition Factors in Metastatic Renal Cell Carcinoma Patients"

_cancers, 2023, doi:10.3390/cancers15235591_

Round 1
Reviewer 1 Report
Comments and Suggestions for Authors This paper entitled “Association of body composition related variables with thera- 2 peutic efficacy of ipilimumab plus nivolumab in patients with 3 metastatic renal cell carcinoma” is interesting prognostic study focusing on body composition parameters in which patients with sarcopenia were associated with lower response rates and shorter survival in patients treated with ipilimumab plus nivolumab when treated with immune checkpoint inhibitors.
I have a few questions for the authors, and I hope they will answer them before the paper is published.
The authors state, "While patients with a high BMI have a higher risk of developing RCC , these patients tend to have a better prognosis , which have been called as “obesity paradox" . Patients with high BMI may be taking various drug therapies to treat underlying diseases such as diabetes mellitus and hypertension. Could it be that the interrelationships with these medications rather affect prognosis?
Does the degree of factors relevant to the diagnosis of sarcopenia (muscle strength, exercise capacity) other than body composition markers, which the authors give as indicators, make a difference in the treatment response rate?
I have the impression that the therapeutic efficacy of immune checkpoint inhibitors for patients with sarcopenia is significantly lower, but is there any difference in side effect or therapeutic tolerability between patients with sarcopenia and those without sarcopenia?
I request that the above questions be addressed in the discussion.
I would like to see better image quality in Figure 2.
Author Response
To reviewer 1,
Thank you for reviewing our paper and for your valuable comments.
- The possible therapeutic effects of antidiabetic drugs and antihypertensive drugs on renal cancer have been well documented. Although there have been no reports of a relationship between the use of these drugs and the therapeutic effects of immune checkpoint inhibitors, there are many reports that they are associated with a better prognosis in patients with renal cancer. These details have been added to the Discussion.
- Although there are a number of articles that have examined the prognostic impact of sarcopenia in patients with advanced cancer, there have been few articles that have evaluated muscle strength and exercise capacity. As we did not collect these data in this study, we were unable to examine the role of muscle strength and exercise capacity. We added this point to the limitation section.
- The relationship between adverse events of immune checkpoint inhibitors and sarcopenia was described using the frequency of irAEs. Although all grade irAEs were higher in the group without sarcopenia, the frequency of Grade ≥ 3 irAEs was independent of sarcopenia. Therefore, sarcopenia is considered to have no significant effect on the tolerability of immune checkpoint inhibitors.
Reviewer 2 Report
Comments and Suggestions for Authors
-

-
Author Response
To reviewer 2,
Thank you for reviewing our paper and for giving us your valuable comments.
First of all, I sincerely apologize that you might have reviewed an older version of the manuscript this time. When I submitted the paper, the editor instructed me to revise it before peer review, and I reposted the revised version. However, an older version seems to have been used for peer review. The paper I am reposting reflects all comments received.
- We added body composition-related biomarker as a keyword.
- The BMI results can be found in the Result section (3.1).
- Women use one cut-off point regardless of BMI.
- A description of T stage has been added.
- We corrected the numerical error.
- We added a description of each graph.
- As described in the result section, there were only normal and overweight patients and no obese cases in this study cohort.
- A detail description of irAE has been added.
9.10. We rephrased “any grade irAE” to “all grade (Grade 1 to 5) irAE”.
- irAEs and antitumor effects have been described separately.
- We added a reference here.
Reviewer 3 Report
Comments and Suggestions for Authors
From a biostats and clinical epidemiology point of view, this is an interesting manuscript and I have some suggestions for the Authors:
- title (and everywhere), use association rather than correlation!
- p-values, to be always exactly reported with 3-sign digits, avoiding generic statements as p<0.01
- for all survival and clinical efficacy outcome measures (OS/PFS/ORR/DCR and so on), it's mandatory to report the exact number of events; even to prove that survival modeling is not overparametrized!
- line 124 The diseafigse control, typo!
- add to stats sections that continuous covariates are reported as median/IQR and categorical ones as absolute/relative frequencies, all the rest is OK!
- line 136 The clinical characteristics of the 60 patients are presented in Table 1: this is not true since you are describing 31 subjects! Thus, this table has to be totally rewritten
- IMHO, two different tables should be prepared, the first with all 60 pts, the second one stratified by the sarcopenia occurrence (y/n); the current table 2 should be omitted
- so, just for now it's more or less difficult to properly evalute all results and conclusions, even being quite interesting
- the vast heterogeneity of ICI+TKI schedules must be clearly underlined
- median follow-up, are you sure that 15 mos are adequate to collect the largest part of events!? I'm not so sure...
- KM curve lack number-at-risk; mind the very low number of pts at any time point!
- Cox models for OS and PFS, please give some details about continuous covariates dichotomization (low vs high?)
- Cox models for OS, the HR over 100 for sarcopenia is a math mistake: this parameter can not be truly estimated, most probably due the extremely low number of sarcopenia-dependent events. In this form, the current Cox OS model can not be accepted
Author Response
To reviewer 3.
Thank you for reviewing our paper and for giving us your valuable comments.
First of all, I sincerely apologize that you might have reviewed the manuscript which includes older version of tables and figures. When I submitted the paper, the editor instructed me to revise it before peer review, and I reposted the revised version. However, it seems that the manuscript you reviewed included older version of tables.
The paper I am reposting includes revised tables and also reflects all comments we received.
We described all p-values with a 3-digit number.
The number of death and progression events is described in the Result section (3.3).
We fixed spelling error according to your suggestion.
As you suggested, we divided Table 1 into Tables 1 and 2, and removed the current table 2.
As you pointed out there are significant differences in drug and dosing schedules between ICI-TKI regimens. As this study included small numbers of patients, we do not intend to investigate difference between regimens. Instead, We described this point in the Limitation section.
We added numbers at risk to KM curves
In the Cox analysis, dichotomization of continuous variables was performed using sex-specific medians, which is described in the method section. We added this explanation to the footnotes of Tables.
Wrong HR values were from older versions of Table. We replaced the old tables with a new Table.
Round 2
Reviewer 3 Report
Comments and Suggestions for Authors
There'a no point to point answer and some concerns do remain unsolved (i.e add to stats sections that continuous covariates are reported as median/IQR and categorical ones as absolute/relative frequencies is still undone)
Comments on the Quality of English Languageminor
Author Response
To reviewer 3,
Firstly, I sincerely apologize for any confusion regarding the version of the manuscript you may have reviewed. Upon submission, the editor requested revisions before peer review, and I subsequently posted the revised version. However, it appears that the manuscript you reviewed included older versions of the tables. The paper I am resubmitting includes the updated tables and incorporates all the comments we have received.
- title (and everywhere), use association rather than correlation!
Response: Your constructive input is greatly appreciated. We have revised the title accordingly.
- p-values, to be always exactly reported with 3-sign digits, avoiding generic statements as p<0.01
Response: We extend our gratitude for your valuable feedback. We have described all p-values in all tables and figures with a 3-digit number.
- for all survival and clinical efficacy outcome measures (OS/PFS/ORR/DCR and so on), it's mandatory to report the exact number of events; even to prove that survival modeling is not overparametrized!
Response: We thank you for your valuable observation. The number of death and progression events is described in the Results section as follows:
During the follow-up period (median, 15 months; range, 1–52 months), 30 (50%) patients developed PD, and 26 (43%) died of the disease.
- line 124 The diseafigse control, typo!
Response: We are grateful for your valuable remark. We have fixed the spelling error.
- add to stats sections that continuous covariates are reported as median/IQR and categorical ones as absolute/relative frequencies, all the rest is OK!
Response: We appreciate your insightful input. We have added a description regarding continuous and categorical variables in the Patients and Methods section.
- - line 136 The clinical characteristics of the 60 patients are presented in Table 1: this is not true since you are describing 31 subjects! Thus, this table has to be totally rewritten
Response: We are grateful for your supportive comment. In our previous submission, we inadvertently included an older version of a table. The revised table we resubmit now contains data from all 60 cases correctly and displays p-values with three digits.
- IMHO, two different tables should be prepared, the first with all 60 pts, the second one stratified by the sarcopenia occurrence (y/n); the current table 2 should be omitted
Response: Your valuable contribution is recognized. Following your suggestion, we have divided Table 1 into Tables 1 and 2, and consequently, removed the previous table 2.
- the vast heterogeneity of ICI+TKI schedules must be clearly underlined
Response: Your informative remark is appreciated. As you pointed out, there are significant differences in drug and dosing schedules between ICI-TKI regimens. Since this study included a small numbers of patients, our intention was not to investigate differences between regimens. Instead, we have described this point in the Limitations section.
- median follow-up, are you sure that 15 mos are adequate to collect the largest part of events!? I'm not so sure...
Response: Your helpful suggestion is noted. As you pointed out, the median follow-up period of 15 months is still short, and a longer observation period is desirable. However, considering that the current study included patients with advanced cancer and has already observed both progression (30) and death (26) events within this follow-up period, we believe it is possible to evaluate survival events. The short observation period was described in the Limitations section.
- KM curve lack number-at-risk; mind the very low number of pts at any time point!
Response: Your insightful feedback is acknowledged. We have added numbers at risk to the K-M curves.
- Cox models for OS and PFS, please give some details about continuous covariates dichotomization (low vs high?)
Response: We appreciate your valuable input. Sarcopenia was precisely defined using sex-specific cut-off values for SMI. For SFI, VFI, and TFI, dichotomization was performed using sex-specific medians as no validated cut-off exists for these variables. This is described in the Patients and Methods section. We have also added this explanation to the footnotes of the tables.
- Cox models for OS, the HR over 100 for sarcopenia is a math mistake: this parameter can not be truly estimated, most probably due the extremely low number of sarcopenia-dependent events. In this form, the current Cox OS model can not be accepted.
Response: Thanks for your beneficial observation. The HR over 100 was from the old table with a small number of patients. We have replaced the old tables with new ones that include data from 60 patients and corrected HR values.
Round 3
Reviewer 3 Report
Comments and Suggestions for Authors
no cover letter, no authors comments, no underlined modifications: this is the worst approach to a multi-stage review process
Comments on the Quality of English Languageminor